# Quality of Life in Female Patients with Overactive Bladder after Botulinum Toxin Treatment

**DOI:** 10.3390/toxins16010007

**Published:** 2023-12-21

**Authors:** Agnieszka A. Licow-Kamińska, Sylwester M. Ciećwież, Magdalena Ptak, Dariusz Kotlęga, Agnieszka Brodowska

**Affiliations:** 1Department of Neonates, Pathology and Intensive Therapy, Independent Public Specialist Institute of Health “Zdroje”, ul. Mączna 4, 70-780 Szczecin, Poland; alicow@interia.pl; 2Department of Children Disease and Children Nursing, Pomeranian Medical University in Szczecin, ul. Żołnierska 48, 71-210 Szczecin, Poland; 3Department of Gynecology, Endocrinology and Gynecologic Oncology, Pomeranian Medical University in Szczecin, ul. Unii Lubelskiej 1, 71-252 Szczecin, Poland; agabrod@wp.pl; 4Independent Subdepartment of Perineological Physiotherapy, Pomeranian Medical University in Szczecin, ul. Żołnierska 54, 51-210 Szczecin, Poland; magdalena.ptak@pum.edu.pl; 5Department of Pharmacology and Toxicology, University of Zielona Góra, ul. Licealna 9, 65-417 Zielona Góra, Poland

**Keywords:** overactive bladder, OAB, urinary incontinence, urinary bladder, botulinum toxin A, botulinum toxin treatment, onabotulinum toxin A, Botox

## Abstract

Background: Manifestations of OAB can considerably diminish the quality of life. Botulinum toxin has emerged as a valuable treatment option in diseases whose symptoms cannot be controlled adequately with other available therapies. The aim of the present study was to compare the subjective quality of life of patients with OAB before the injection of botulinum toxin and three and six months after the intervention. Methods: This study was based on a diagnostic survey with three validated questionnaires, ICIQ-OAB, ICIQ-OABqol, and ICIQ-LUTSqol, and an additional questionnaire developed by the authors to collect sociodemographic characteristics and selected medical data. Results: This study demonstrated significant differences between pre-treatment scores and those at three and six months post injection. At three and six months after the intervention, mean scores for all three instruments (ICIQ-OAB, ICIQ-OABqol, ICIQ-LUTSqol) were significantly lower than the respective pre-treatment values, implying a significant attenuation of OAB symptoms and their lower impact on the quality of life. However, the severity of OAB symptoms and their impact on the quality of life at six months post intervention were significantly higher than at three months, except for the social interaction domain. Conclusions: Botulinum toxin is an effective treatment for OAB.

## 1. Introduction

Quality of life (QoL) is a versatile term referring to all factors that affect the life of an individual. The term has been discussed in medical literature since as early as the 1960s [1]. The growing interest in QoL-related issues is associated with the emergence of a new medical concept, promoting a holistic approach to the patient [2]. The multidimensional unfavorable effect of overactive bladder (OAB) manifestations on the QoL contributes to further deterioration in the patient’s health. Understanding one’s health condition or QoL is essential for evaluating the effectiveness of new treatments. The diagnosis of OAB is based primarily on patient-reported manifestations, such as urgency with or without urge incontinence, usually increased frequency of passing urine, and nocturia. These manifestations substantially affect the QoL of millions of patients with OAB [3]. Individuals with OAB often abandon many aspects of their lifestyle, which has a detrimental effect on their social interactions, professional careers, interpersonal and sexual relationships, and psychological well-being [4,5,6]. Botulinum toxin, previously considered a lethal poison, is nowadays recognized as a medication in multiple medical disciplines. The discovery of botulinum toxin and its transition from a life-threatening biological compound to a medication is unusual and fascinating. Successful research by dr Alan Scott led to the United States Food and Drug Administration (FDA) approving botulinum toxin for the treatment of strabismus, blepharospasm, and hemifacial spasm in 1989. Since then, clinical applications of BoNT have expanded in various medical fields, and in 2011, the FDA approved botulinum toxin type A for the treatment of neurogenic overactive displacement (nDO), and further approved it in 2013 for the treatment of refractory OAB. Botulinum toxin is a neurotoxin synthesized by a bacterium, Clostridium botulinum. BoNT is classified into seven distinct neurotoxins (i.e., types A–G) that inhibit acetylcholine release at the pre-synaptic cholinergic neuromuscular junction to paralyze muscles. Blocking the release of acetylcholine inhibits parasympathetic signaling to the bladder, reducing involuntary detrusor contractions. Two serotypes, type A (BoNT-A) and type B (BoNT-B), are currently in clinical use. The primary treatment of OAB is behavioral therapies and lifestyle modifications, followed by pharmacotherapy (antimuscarinic drugs and/or beta-3 adrenergic agonist) as a second-line therapy. However, oral medications fail to provide effective treatment and have poor long-term tolerance. In patients refractory to these OAB medications, intravesical Botox injection has been documented to act as a third-line treatment according to the American Urological Association (AUA) and European Association of Urology guidelines. Botulinum toxin injections can be an alternative therapy in the course of diseases resistant to standard treatment [7]. Social awareness of and demand for this treatment modality are constantly increasing, expanding the activities of midwives, physicians and entire medical teams, who consider the full or partial recovery of their patients a successful outcome. The primary aim of this study was to compare the subjective QoL of female patients with OAB before botulinum toxin treatment and three and six months after the intravesical injection of that agent.

## 2. Results

### 2.1. Descriptive Statistics

Before verifying the study hypotheses, basic descriptive statistics were calculated for each variable. Those statistics are presented in Table 1, Table 2 and Table 3, along with the results of the respective statistical tests, separately for measurement 1 [P1] taken before botulinum toxin treatment and measurements 2 [P2] and 3 [P3] obtained three and six months post treatment, respectively.

The last question of the ICIQ-LUTSqol questionnaire is not included in the domains mentioned above and refers to the degree to which OAB symptoms interfere with everyday life. The impact of the symptoms is scored on a scale from 0 to 10. The results are presented in Table 4.

### 2.2. Differences between QoL Scores in Domains Associated with OAB Symptoms

The differences in the pre- and post-treatment ICIQ-OAB scores in specific domains, i.e., frequency of passing urine, nocturia, urgency, urge incontinence, and overall score for the questionnaire, turned out to be statistically significant. The effect sizes, estimated based on eta-square values, were very high, explaining 58% to 74% of the variance in the results.

Subsequently, post hoc tests were carried out with the Sidak correction for multiple comparisons. The results of post hoc testing for all variables mentioned above followed the same pattern of between-measurement differences. The pre-treatment values differed significantly from the results obtained three and six months after treatment, with mean results for both post-treatment time points being significantly lower than at baseline. This implies that OAB symptoms attenuated significantly 3 and 6 months after botulinum toxin injection. However, the severity of symptoms at six months was significantly higher than at three months, as shown by higher mean values for all analyzed variables. The results are presented in detail in Table 5 and Table 6.

Analogous results were obtained when a similar analysis was carried out for the ICIQ-OABqol scores. The results of consecutive measurements for each domain of the instrument differed significantly, with the effect sizes explaining 57% to 66% of the variance in the results, which corresponded to substantial between-measurement differences.

The results of post hoc tests with the Sidak correction for all parameters except the scores for the “Social Interaction” domain followed the same pattern as in the case of the ICIQ-OAB questionnaire. Thus, the results obtained at three and six months post treatment were significantly lower than at baseline, whereas scores at six months were significantly higher than those at three months. In the case of the “Social Interaction” domain, scores at six months did not differ significantly from those at three months. The results are presented in detail in Table 7 and Table 8.

Similar results were also obtained for the scores of the ICIQ-LUTSqol questionnaire. The analysis of variance demonstrated that consecutive scores for all analyzed variables differed significantly from one another, with very high effect sizes explaining 48% to 66% of variance in the results.

The results of post hoc testing demonstrated that the scores of the ICIQ-LUTSquol followed the same pattern as the scores for all domains of the ICIQ-OAB and most domains of the ICIQ-OABqol instrument. “Social limitations” was the only domain of the ICIQ-LUTSquol, for which the results obtained at six months differed significantly from those recorded at three months. The results are presented in detail in Table 9 and Table 10.

Also, the differences in the pre- and post-treatment impact of OAB symptoms on everyday life were tested with repeated measures ANOVA. The analysis demonstrated that the treatment effect explained up to 71% of the variance in the results. The impact of OAB symptoms on everyday life was the highest before treatment, then reached its lowest level at three months and again increased significantly by month 6. The results are presented in detail in Table 11 and Table 12.

### 2.3. Impact of OAB Symptoms on Various Aspects of Life

Each item of the ICIQ-OAB questionnaire included a question about the impact of a given OAB symptom, scored on a scale from 0 (not at all) to 10 (a great deal). This scale describes how much the patient is bothered by a given symptom. A similar question, again scored from 0 (not at all) to 10 (a great deal), was also included in every item of the ICIQ-LUTSqol questionnaire. A series of Friedman ANOVA tests were conducted to identify potential differences between the three consecutive measurements of OAB symptom impact. The results of all four analyses turned out to be statistically significant (*p* < 0.05), and the effect sizes were very high (*W* > 0.8). The results of post hoc tests with the Bonferroni correction revealed a shared pattern for all OAB symptoms, i.e., the frequency of passing urine, nocturia, urgency, and urge incontinence. Regardless of the symptom, impact at baseline differed significantly from that recorded at three and six months after botulinum toxin injection. The impact of all OAB symptoms before treatment was significantly higher than after, with no significant difference between values documented at three and six months after the intervention (Table 13 and Table 14).

Analogous analyses were also conducted for the impact of OAB on various aspects of life, determined with the ICIQ-LUTSqol instrument. The results of all Friedman tests were statistically significant, with very high effect sizes. The results of post hoc testing demonstrated that the impact of OAB on all aspects of life except sleep quality followed the same pattern as described above for specific urinary symptoms. Thus, the impact of the disease on various aspects of life at three and six months post intervention was significantly lower than before botulinum toxin injection, with no significant differences observed between the measurements taken at months 3 and 6. Meanwhile, the pre-treatment impact of OAB on sleep quality decreased significantly at months 3 and 6, yet the impact at month 6 was slightly, albeit significantly, higher than at month 3 (Table 15 and Table 16).

## 3. Discussion

Research on the QoL in chronic disorders is a valuable source of medical information. First, such studies present the illness from the patient’s viewpoint, which can differ from that of healthcare providers, and such subjective assessments of patients are an accessible source of information that might be crucial when making therapeutic decisions. Moreover, research on the QoL may reveal, previously unnoticed by the therapeutic team, the needs of patients associated with providing professional healthcare outside the hospital. Treatment of OAB is aimed at relieving symptoms, not changing the pathological condition. With the development of pharmacotherapy and the discovery of pathophysiology, newer treatments have been shown to have greater efficacy and fewer side effects [8].

Oral pharmacotherapy with antimuscarinic agents and/or β3-agonists is a standard of care in patients with OAB, but this treatment can be ineffective in some cases, and some patients do not respond to the therapy. Additionally, it is not infrequent for these drugs to have limited effectiveness and/or be tolerated poorly, which results in a lack of medication adherence and pre-term discontinuation [9]. According to the literature, more than half of OAB patients are not satisfied with anticholinergic treatment [10]. Botulinum toxin injections are the third line of treatment in patients with OAB in whom oral pharmacotherapy turned out to be ineffective. Due to its biological properties, botulinum toxin may constitute an alternative treatment for an array of diseases that are unresponsive to conventional therapy. In the present study, pharmacotherapy failed to produce the desired effect in all the patients. Meanwhile, the injection of botulinum toxin A resulted in a substantial improvement in QoL. The improvement persisted throughout the entire six-month study period. These findings are consistent with the results of other published studies analyzing the effects of botulinum toxin treatment in OAB [11,12,13].

In this study, the highest baseline scores were recorded for urgency, which suggests that this symptom was reported by the study patients particularly often. Not only did the occurrence of urgency episodes decrease with time, but other OAB symptoms were also reported less often. These beneficial effects were associated with improved QoL measured with the ICIQ-OABqol and ICIQ-LUTSqol scales. This improvement was observed in each aspect of life, as shown by significant changes in scores for all domains of those QoL instruments.

Previously published randomized controlled trials compared outcomes in patients receiving botulinum toxin injections and those given placebo. Importantly, those studies showed significant differences between the two groups, suggesting that botulinum toxin injections contributed to attenuating OAB symptoms. One limitation of the present study may be the lack of a reference group. Nevertheless, all patients included in the present study received botulinum toxin, and both our findings and the evidence from randomized controlled trials show that this treatment is an effective option in the management of OAB. However, we still know little about the long-term outcomes of the treatment, as even in the studies with the longest follow-up period, patients were monitored for no longer than 12 weeks after botulinum injection [14]. Moreover, it must be emphasized that many published studies analyzing the QoL in OAB after botulinum toxin A (BoNT/A) injection included both female and male patients, and therefore, comparison between our findings and their results requires some caution.

The results of the present study are worth emphasizing, given the paucity of pharmacotherapy options for patients who do not respond adequately to anticholinergic agents.

### 3.1. Effect of Botulin Toxin Treatment on the Attenuation of OAB Symptoms

The analysis of ICIQ-OAB scores demonstrated a significant decrease in the frequency of OAB manifestations, namely, the frequency of passing urine, nocturia, urgency, and urge incontinence, both at three and six months after the intravesical injection of botulinum toxin A. Our findings are consistent with the results published by Juszczak et al. [13], who also observed a significant decrease in the severity of OAB symptoms after administering botulinum toxin A. According to those authors, the attenuation of symptoms was observed for up to nine months post-injection, and then, the severity increased, albeit insignificantly, at month 9. Similarly, in our study, a non-significant increase in symptom severity was observed at six months after the BoNT/A injection, compared with that recorded at month 3. Nevertheless, the impact of OAB symptoms before the treatment was significantly higher than at months 3 and 6, with no significant differences observed between the latter two study time points.

In the present study, the frequency of passing urine decreased significantly at three and six months after treatment (from a mean value of 69.75 before the treatment to 20 at month 3 and 29.50 at month 6, *p* < 0.001). These observations are consistent with the results of a prospective multinational study conducted in Germany, Spain, Sweden, and the United Kingdom in a cohort of patients with OAB and urinary incontinence (UI). The authors of that study, Hamid et al. [15], demonstrated a decrease in the frequency of passing urine at one and twelve weeks after botulinum toxin A injection (from 11.2 at baseline to 8.9 at week 1 and 7.7 at week 12). While the results presented above are cumulative values for female and male patients, up to 84.9% of the respondents were women. A decrease in the frequency of passing urine was also observed in randomized controlled trials comparing the outcomes of patients with OAB receiving botulinum toxin injections and placebo. According to Chapple et al. [16], onabotulinum toxin A treatment (100 units) was associated with a clinically relevant reduction in urinary symptoms in patients with OAB and UI who previously did not respond adequately to anticholinergic treatment. More than 80% of the study group, 88.1% in the BoNT/A arm and 84.5% in the placebo arm, were women. At week 12 of the study, the severity of all OAB symptoms in patients who received onabotulinum toxin A was significantly lower than in the placebo group (*p* < 0.001 for all symptoms except nocturia, *p* < 0.01 for nocturia). Similar findings were reported by Nitti et al. [17], who observed a decrease in the mean frequency of passing urine at twelve weeks after botulinum toxin injection (−2.15 vs. −0.91 in the placebo arm, *p* < 0.001). In that study, too, women constituted most of the study group, 90% in the BoNT/A arm and 84.4% in the placebo arm. Yokoyama et al. [18] also analyzed the effectiveness of botulinum toxin A at week 12 post injection in Japanese patients with OAB and UI. The study group consisted of men and women (74% of women in the BoNT/A arm and 76% in the placebo arm). The mean decrease in the frequency of passing urine in patients who received botulinum toxin injection was significantly higher than in those treated with placebo (−1.87 vs. −0.42, *p* < 0.001). Moreover, the study demonstrated that botulinum toxin treatment contributed to a significant decrease in the overall severity of OAB symptoms expressed as the Overactive Bladder Symptom Score (OABSS). The OABSS was also used as an outcome measure by Wang et al. [19], who documented a significant attenuation of OAB symptoms in Taiwanese patients during a 12-week follow-up after treatment with botulinum toxin type A. The latter study included 153 women and 62 men with OAB.

It is important that botulinum toxin dosages are expressed in units of biological activity, which are different for each product. This means that the units used in the dosage of one product cannot be used for other botulinum toxin products. In China, botulinum toxin A is marketed under the trade name Hengli^®^ (Lanzhou Biological Products Institute, Lanzhou, China). According to Liao et al. [20], six weeks after the injection of Hengli^®^ (100 units), a reduction in the mean number of urine passing episodes per day in the active treatment group was significantly higher than in the placebo group (−2.40 vs. 0.70; *p* = 0.003). Most of the study group, 82.39% in the BoNT/A arm and 85.92% in the placebo arm, were women.

Tamburro et al. [21] analyzed the outcomes of botulinum toxin treatment in women and men with OAB unresponsive to anticholinergic agents. Most of the study group were women (77%). The study identified the frequency of passing urine as the symptom in the case of which botulinum toxin treatment contributed to the most pronounced reduction at week 12 (from 11.3 at baseline to 5.8 at week 12). Meanwhile, in our present study, the most evident effect of botulinum toxin treatment was observed in the case of urgency.

Miotła et al. [12] analyzed the effect of botulinum toxin A on the frequency of nocturia in 76 women with OAB during a 12-week follow-up. According to those authors, BoNT/A injections effectively reduced the frequency of nocturia (mean reduction −0.98; *p* < 0.001) and night episodes of urge urinary incontinence (UUI, mean reduction −0.37; *p* < 0.001). Our present study also demonstrated a significant post-treatment decrease in the frequency of nocturia (from 75.75 at baseline to 34.25 at month 3 and 41.50 at month 6, *p* < 0.001). These findings are consistent with the results of other studies. Tamburro et al. [21] observed a decrease in nocturia frequency during a 12-week follow-up (from 2.7 at baseline to 0.7 at week 12). A similar trend was also reported by Hamid et al. [15] (a decrease from 2.6 at baseline to 1.8 at week 1 and 1.2 at week 12). Their observations are in agreement with the results published by Chapple et al. [16] and Nitti et al. [17], according to whom a decrease in nocturia frequency at week 12 post intervention was significantly higher in patients who received botulinum toxin than in those receiving placebo (mean decrease: −0.54 vs. −0.25 *p* < 0.01 and −0.45 vs. −0.24, *p* ≤ 0.05, respectively). A similar relationship was also observed in another randomized trial conducted by Yokoyama et al. [18] (mean decrease at week 12: −0.30 vs. 0.03, *p* = 0.048). Interestingly, however, the latter study did not demonstrate a significant between-group difference in the occurrence of nocturia at week 2 post intervention (mean decrease −0.10 vs. −0.05, *p* = 0.751).

The present study identified urgency as the most common manifestation and most bothersome symptom reported by women with OAB, which is consistent with the results published by Coyne et al. [22]. Similar to the present study, Hamid et al. [15] also observed a significant decrease in the occurrence of urgency at 12 weeks after the intravesical injection of botulinum toxin (7.6 at baseline vs. 4.2 at week 1 vs. 2.5 at week 12). According to Miotła et al. [12], up to 69.7% of female patients reported a ≥50% improvement regarding urgency at 12 weeks after BoNT/A injection. Our present study also demonstrated a decrease in urgency, from 85.00 at baseline to 31.50 at month 3 and 40.25 at month 12, with the values at both latter time points being significantly lower than at baseline (*p* < 0.001). Additionally, urgency was the symptom with the highest absolute reduction among all OAB manifestations (a decrease of 44.75 at month 6).

Randomized studies conducted by Chapple et al. [16] and Nitti et al. [17] also documented a significant (*p* < 0.001) decrease in urgency at week 12 after the administration of onabotulinum toxin A. According to Yokoyama et al. [18], at 12 weeks post intervention, the decrease in the number of urgency episodes in the botulinum toxin arm was significantly higher than in the placebo arm (−3.40 vs. −1.17, *p* < 0.001). However, the between-group difference did not reach the statistical significance threshold at week 2 post intervention (mean decrease −2.10 vs. −1.20, *p* = 0.112). More evidence of improvement in the mean daily number of urgency episodes originates from a study conducted by Sievert et al. [23]. In that study, which included up to 87.8% of female patients, the mean decrease in the number of urgency episodes at 12 weeks post intervention was significantly higher in the botulinum toxin arm than in the placebo arm (3.30 vs. 1.23; *p* < 0.001). Also, in a Chinese study, the administration of Hengli^®^ (100 units) contributed to a significant decrease in urgency episodes at weeks 2, 6, and 12 compared with placebo [20].

According to many authors, botulinum toxin injection results in a lower frequency of UUI. Interestingly, Hamid et al. [15] demonstrated that in 25.5% of the patients, UI resolved completely (i.e., the number of UI episodes decreased by 100%) within one week after botulinum toxin injection, and then, this percentage increased up to 41.8% at week 12. Additionally, the daily frequency of UI episodes decreased by at least 50% in 60.7% and 73.9% of the patients at weeks 1 and 12, respectively. Tamburro et al. [21] also reported a significant decrease in the daily number of UUI episodes during a 12-week follow-up, from 3.6 at baseline to 1 at week 12. According to Hendrickson et al. [24], the median time to relapse of UUI in women with neurogenic UUI treated with botulinum toxin injections (100 or 200 units) was six months. Meanwhile, in the present study, the occurrence of UUI at six months after injecting 100 units of BoNT/A was still significantly lower than before the treatment.

Consistently with the findings reported above, randomized trials also demonstrated that botulinum toxin injections contributed to a decrease in the number of UUI episodes in patients with OAB. According to Chapple et al. [16], the mean reduction in the number of UUI episodes at week 12 post intervention was significantly higher in the onabotulinum toxin A group than in the placebo group (−2.95 vs. −1.03; *p* < 0.001). Those findings are consistent with the results published by Nitti et al. [17], according to whom BoNT/A injections contributed to a significant decrease in the number of UUI episodes at week 12 (mean decrease of −2.65 as compared with −0.87 for placebo; *p* < 0.001). Moreover, up to 57.7% of the patients presented with at least a 50% reduction in UI at week 12 and 22.09% achieved complete continence (i.e., a 100% reduction in UI episodes). According to Yokoyama et al. [18], a mean decrease in the baseline number of UI episodes in the onabotulinum toxin A arm was significantly higher than in the placebo arm (−3.42 vs. −1.25; *p* < 0.001). The significant between-group difference observed during the first assessment at week 2 post-intervention persisted until the last follow-up visit at week 12 (mean reduction −3.13 vs. −1.02; *p* < 0.001). The results of the three randomized trials mentioned above, conducted in Europe, North America, and Japan, were consistent regarding the mean reduction in UUI episodes at 12 weeks after botulinum toxin injection. The results of our present study are on par with those findings, as botulinum toxin treatment also contributed to a significant decrease in the number of UUI episodes, from 63.25 at baseline to 21.75 at month 3 and 28.75 at month 6 (*p* < 0.001).

Notably, Amundsen et al. [25] demonstrated that the mean reduction in the number of UUI episodes in women treated with botulinum toxin A was significantly higher than in those subjected to sacral neuromodulation.

### 3.2. Effect of Botulinum Toxin Treatment on Improvement in QoL

In the present study, QoL after botulinum toxin treatment was assessed with two instruments, ICIQ-OABqol and ICIQ-LUTSqol. These instruments comprehensively analyze the effect of OAB manifestations on patients’ lives. Before treatment with botulinum toxin A, the QoL of the study participants was low, as shown by the high scores obtained with both scales. Analyzing the treatment outcomes, we focused on QoL in various domains. The lower the score for a given domain after the treatment, the lesser the impact of OAB symptoms on this sphere of the patient’s life. Additionally, we calculated the overall scores for both scales. A lesser impact of OAB symptoms after the treatment, demonstrated with the ICIQ-OAB scale, reflected a better QoL of the patients, as shown by a significant post-treatment improvement in specific domains of the ICIQ-OABqol and ICIQ-LUTSqol. A lesser impact of OAB manifestations on QoL was also documented based on significant changes in overall ICIQ-OABqol scores (a decrease from 246.78 at baseline to 91.07 and month 3 and 113.32 at month 6, *p* < 0.001) and overall ICIQ-LUTSqol scores (a decrease from 535.75 at baseline to 201.61 at month 3 and 254.69 at month 6, *p* < 0.001). A significantly lesser impact of symptoms on QoL was also demonstrated in all specific domains. The improvement in QoL documented herein is on par with the results of previous studies dealing with the problem in question. Miotła et al. [12] demonstrated a significant improvement in all domains of the King’s Health Questionnaire (KHQ) during a 12-week follow-up after treatment with BoNT/A (*p* < 0.001). A considerable improvement in QoL, measured at 12 weeks post-treatment with the 36-Item Short-Form Health Survey, was also observed in a prospective study conducted by Tamburro et al. [21], along with a significant decrease in symptom severity determined with the Overactive Bladder Screener, from 34.5 to 17.1 (*p* < 0.05). Those findings were consistent with the results obtained with the Treatment Benefit Scale (TBS), according to which up to 87% of the patients perceived a considerable improvement in their health status. The subjective opinions of patients about their health status expressed with the TBS were also considered in several randomized trials. The proportions of patients who perceived a considerable improvement in their health status (TBS scores) after treatment with BoNT/A in the studies conducted by Chapple et al. [16] and Nitti et al. [17] were similar, 62.8% and 60.8%, respectively. A significant post-treatment improvement in TBS scores was also reported by Yokoyama et al. [18] and Sievert et al. [23].

The present study showed that the overall impact of OAB manifestations on QoL increased slightly at six months after botulinum toxin treatment. A similar relationship was reported previously by Juszczak et al. [13], who observed a slightly higher impact of urinary ailments on QoL at six and nine months after botulinum toxin injection. The latter authors analyzed the effectiveness of botulinum toxin A in treating pharmacotherapy-resistant OAB by assessing KHQ scores at 3, 6, and 9 months after the intervention. Although, in the present study, the overall impact of OAB manifestations on QoL increased slightly at six months post-intervention, the impact at both three and six months was still significantly lower than at baseline. Nevertheless, the impact of OAB symptoms on QoL measured with both scales (except “Social Interaction” and “Social Limitations” assessed with ICIQ-OABqol and ICIQ-LUTSqol, respectively) increased significantly when the results recorded at six months were compared with those obtained at three months. Additionally, we used the ICIQ-LUTSqol to analyze the burden the patients experienced in various aspects of their lives. Regardless of the analyzed aspect of life, the burden at three and six months after treatment was significantly lower than at baseline. Meanwhile, no significant differences were found between the scores for months 3 and 6, except for a slight increase in sleep quality scores.

The impact of OAB symptoms on everyday life is not easy to determine as the QoL in this domain is modulated by various factors. Those factors are included, among others, in the “Copying” domain of the ICIQ-OABqol questionnaire. This domain contains questions about journey planning, avoiding activities associated with restricted access to a toilet, and decreasing physical activity. Our study demonstrated that the impact of OAB symptoms on those activities was significantly reduced three and six months after the intervention, from 72.73 at baseline to 27.58 at month 3 and 33.85 at month 6 (*p* < 0.001). Six months after the intervention, the absolute change in “Copying” was the highest of all the ICIQ-OABqol domains, with a difference of 38.88. ICIQ-LUTSqol measures the impact of OAB symptoms on everyday life in the domains “Role Limitations”, covering such activities as cleaning, shopping, etc., and “Physical Limitations”, defined as going for a walk, running, doing sport, going to the gym, etc., ability to travel with a car, bus, train, plane, etc. Our analysis demonstrated a significant improvement in the QoL in those domains (“Role Limitations”: from 80.50 at baseline to 29.33 at month 3 and 37.67 at month 6, *p* < 0.001; “Physical Limitations”: from 83.50 at baseline to 31.50 at month 3 and 39.33 at month 6, *p* < 0.001). Six months after the intervention, the absolute change in “Physical Limitations” was the highest of all the ICIQ-LUTSqol domains, with a difference of 44.17. In many previous studies, the impact of OAB symptoms on the QoL in everyday life was assessed with the KHQ. ICIQ-LUTSqol is an adaptation of the KHQ, designed to be used as a part of the ICIQ; thus, most questions included in those two questionnaires overlap. The KHQ contains a question about the impact of urinary ailments on everyday life, along with the “Role Limitations” domain, referring to the impact of the ailments on household tasks (e.g., cleaning) and daily activities outside the home, such as shopping, work, etc., and the “Physical Limitation” domain, measuring limitations in physical activities (going for a walk, doing sport, etc.) and in the ability to travel. The KHQ has been used by several authors, including Rechberger et al. [11], who demonstrated a significant decrease in the impact of OAB symptoms on those aspects of life three months after botulinum toxin injection. Those findings are consistent with the results reported by Juszczak et al. [13], who also used the KHQ. According to those latter authors, the impact of OAB symptoms on activities of daily living (both household tasks and activities outside the home) and physical activity at 3, 6, and 9 months after botulinum toxin injection was significantly lower than before the treatment. The same study did not show a significant effect of the treatment on scores for the “General Perception of Health” domain; the scores recorded at 3, 6, and 9 months post treatment did not differ significantly from the pre-treatment values. These observations are in agreement with the results published by Chapple et al. [16], who also did not find a significant impact of BoNT/A treatment on that domain at 12 weeks post intervention. The last question of the ICIQ-LUTSqol questionnaire used in the present study referred to the degree to which urinary symptoms interfered with the everyday lives of the respondents. Our analysis showed that the impact of OAB symptoms on the everyday life of the respondents had decreased significantly after botulinum toxin treatment. The impact of OAB symptoms was shown to be the lowest at three months and then increased significantly by month 6.

In the study conducted by Chapple et al. [16], the scores for the “Role Limitations” domain were the highest of all the KHQ domains, which implies that OAB symptoms had a considerable impact on the QoL in that area. The study documented a significant post-treatment decrease in the scores for the “Role Limitations” and “Physical Limitation” domains, corresponding to an improvement in QoL in those aspects of life. The same study assessed the QoL with another instrument, the Incontinence Quality of Life (I-QOL) questionnaire. The instrument contains three domains: “Avoid and Limiting Behaviors”, “Psychosocial Impacts”, and “Social Embarrassment”. According to Souza et al. [26], the “Social Embarrassment” domain of the I-QOL is an equivalent of the “Role Limitations”, “Physical Limitation”, “Social Limitation”, and “Emotions” domains of the KHQ, whereas the “Psychosocial Impacts” domain corresponds well with the “Social Limitation” and “Emotions” domains. Chapple et al. [16] observed a significant decrease in the impact of urinary symptoms on the “Social Embarrassment” and “Psychosocial Impacts” domains of the I-QOL. These findings are consistent with the results published by Nitti et al. [17], who used the I-QOL and KHQ to document a significant decrease in the impact of OAB symptoms on the same aspects of life. The I-QOL has also been used by Fowler et al. [27] in a randomized study comparing the effects of botulinum toxin injections (50, 100, 150, 200, 300 units) with placebo. Women constituted up to 92.0% of the study participants. The study showed that treatment with onabotulinum toxin A (≥100 units) contributed to a significant improvement in the overall I-QOL score and scores for all its domains during each follow-up visit between weeks 2 and 24 (*p* < 0.05). In line with these findings, Yokoyama et al. [18] observed a significant improvement in the QoL in the “Role Limitations” and “Social Limitation” domains at 12 weeks after the injection of BoNT/A.

Ginsberg et al. [28] reported the outcomes of repeated treatment with botulinum toxin in patients with OAB and UI. Most of the patients were women (90.3%). The study documented a permanent improvement in QoL regarding job and everyday activities. Between the first and the sixth round of treatment, an improvement in those domains was reported by 56% to 72% of the respondents.

Detailed analysis of the data demonstrated that social interactions were the only aspect of life in the case of which mean scores determined at six months were not significantly higher than those recorded at three months. This suggests that solely in this domain, the post-treatment improvement persisted at a similar level at both three and six months post intervention. This phenomenon was observed both in the case of the “Social Interaction” domain of the ICIQ-OABquol questionnaire (mean baseline score: 42.88, score at month 3: 14.48, score at month 6: 17.32, *p* < 0.001) and the “Social Limitations” domain of the ICIQ-LUTSquol questionnaire (mean baseline score: 54.83, score at month 3: 16.83, score at month 6: 21.00, *p* < 0.001).

Relations with others make our lives more satisfactory. OAB symptoms undoubtedly impact contact with others and social life. The present study demonstrated that the impact of OAB symptoms on social interactions decreased significantly after botulinum toxin treatment. A similar phenomenon was previously reported by Rechberger et al. [11] and Juszczak et al. [13]. The results of the present study are also in agreement with evidence from randomized trials that documented a significant improvement in the “Social Limitation” domain of the KHQ at 12 weeks after the injection of botulinum toxin A [16,17,18].

Stress and negative emotions associated with OAB symptoms may have a substantial impact on the quality of family relationships. Also, relationships with intimate partners are, without a doubt, of utmost importance. Sex life is an integral element of the QoL and can be significantly affected by urinary ailments. Moreover, the quality of the intimate relationship with a spouse/partner is a significant determinant of other family relationships. The “Personal Relationships” domain of the ICIQ-LUTSqol questionnaire extends to close relationships with others, with a particular emphasis placed on the relationship with the intimate partner and harmonious relationships with other family members. In the present study, the impact of OAB on family relationships was analyzed in a group of 90 patients, as the other ten respondents were living in single households. Our analysis demonstrated that intravesical injections of botulinum toxin contributed to a significant decrease in the impact of OAB symptoms on the “Personal Relationships” domain (mean baseline score: 51.36, score at month 3: 16.17, score at month 6: 21.06, *p* < 0.001). These findings are consistent with the results published by Juszczak et al. [13], who observed a similar relationship at three and six months post injection but found no significant improvement at nine months. Rechberger et al. [11] reported an improvement in the “Personal Relationships” domain of the KHQ, with mean scores three months after BoNT/A injection being lower, albeit not significantly, than the pre-treatment values. A post-treatment improvement in “Personal Relationships” scores was also previously documented by Chapple et al. [16] and Nitti et al. [17].

OAB symptoms can be detrimental to the quality of sex life. According to the literature, UI is associated with a higher incidence and greater severity of sexual dysfunction in female patients [29]. Balzarro et al. [30] analyzed the effect of onabotulinum toxin injection on the improvement of sexual function in women with wet OAB. Sexual function was assessed using the Female Sexual Function Index (FSFI) scale, showing a post-treatment improvement in most domains except “Desire” and “Pain”. A significant change in overall FSFI scores (*p* = 0.0008) suggested that the injection of onabotulinum toxin A (100 units) contributed to an improvement in sexual function. Additionally, the authors found a significant correlation between a decrease in the number of UUI episodes and an improvement in the overall FSFI score, demonstrating that UUI had the strongest impact on sexual function in the study population. FSFI was also used by Miotła et al. [31], who analyzed the effect of botulinum toxin treatment on sexual function in women with OAB. Before the treatment, the study participants showed a lower quality of sexual function in all FSFI domains than healthy women (median overall score 21.8 vs. 26.3, *p* < 0.001). At 12 weeks after the treatment, a significant improvement was observed in all FSFI domains. Moreover, over 90% of the respondents showed a clinically relevant increase in the overall score from the baseline values. Botulinum toxin injection contributed to a considerable improvement in sexual function in OAB patients, with post-treatment scores in two domains, “Climax” and “Pain”, reaching the levels documented in healthy women. A beneficial effect of botulinum toxin treatment on sexual function in patients with OAB was also documented in a systematic review published by Shawer et al. [32]. Meanwhile, Ginsberg et al. [28] demonstrated a permanent improvement in the quality of sexual life after repeated BoNT/A injections. After up to six cycles of the treatment, an improvement was reported by 74–78% of the patients. In our present study, the negative impact of OAB symptoms on sexual life at both three and six months after the intervention was significantly lower than before the treatment, with no significant differences found between the results for months 3 and 6. Similar tendencies were also documented for the impact of OAB symptoms on personal relationships and family life.

Sleep is an essential component of somatic and psychological health. Nocturia disrupts sleeping, which may lead to fatigue, sleepiness, lack of concentration, mood swings, lesser efficiency, cognitive function impairment, and higher levels of stress. Our present study and previous research confirmed that treatment with botulinum toxin results in a lesser occurrence of nocturia and, as a result, contributes to better sleep quality. Our analysis demonstrated a significant decrease in the impact of OAB symptoms on the “Sleep” domain of the ICIQ-OABqol (mean baseline score: 65.32, score at month 3: 25.96, score at month 6: 32.92, *p* < 0.001) and the “Sleep/Energy” domain of the ICIQ-LUTSqol (mean baseline score: 69.83, score at month 3: 28.83, score at month 6: 37.33, *p* < 0.001). These findings are consistent with results published by Rechberger et al. [11], Juszczak et al. [13], and Miotła et al. [12]. Previous randomized studies also showed a considerable improvement in scores for the “Sleep/Energy” domain of the KHQ at 12 weeks after the injection of botulinum toxin A. Such an improvement was reported by Chapple et al. [16].

As mentioned above, botulinum toxin injections contributed to an improvement in the quality of sleep in the responders. However, an analysis of the results obtained with the ICIQ-LUTSqol questionnaire showed that the quality of sleep was the only analyzed aspect of life in which the unfavorable impact of OAB symptoms increased slightly at 6 months compared with month 3 post intervention.

OAB symptoms are particularly often associated with shame, embarrassment, fear, anxiety, and depression, which have a profound impact on patient well-being and satisfaction with life. The “Concern” domain of the ICIQ-OABqol refers to many emotions, including distress, frustration, embarrassment, and the feeling of something being wrong. Our analysis showed that all those negative emotions diminished three and six months post intervention (mean baseline score: 65.86, score at month 3: 23.06, score at month 6: 29.23, *p* < 0.001). A beneficial effect of treatment with BoNT/A on the emotions of patients was also previously reported by Rechberger et al. [11] and Juszczak et al. [13]. Meanwhile, Tamburro et al. [21] observed the most evident post-treatment improvement in the domain “Role Limitations Due to Emotional Problems” (items 19 to 82) of the 36-item Short Form Health Survey (SF-36). The SF-36 is an instrument for evaluating health-related quality of life (HR-QoL). The instrument measures QoL in eight dimensions covering different aspects of life. The scores for each dimension are summed up and transformed onto a scale from 0 (the worst QoL) to 100 (the best QoL). Notably, at the end of the present study, at month 6, the most evident improvements were observed in the “Copying” domain of the ICIQ-OABqol and the “Physical Limitations” domain of the ICIQ-LUTSqol instrument. Interestingly, Fowler et al. [27], whose study was based on the SF-36, observed that in patients with OAB, baseline scores in many domains of that instrument were similar to those recorded in patients with diseases causing chronic disability, such as multiple sclerosis and osteoarthritis. Those findings are proof of the negative impact of OAB symptoms on QoL. Our present study also showed a significant post-treatment improvement in QoL in the “Emotions” domain of the ICIQ-LUTSqol (mean baseline score: 60.44, score at month 3: 20.89, score at month 6: 27.67, *p* < 0.001); this domain refers to such emotions as depression, feeling anxious or nervous, and feeling bad about oneself. The instrument also includes the “Embarrassment” domain, the scores for which also improved significantly after treatment (mean baseline score: 67.33, score at month 3: 25.00, score at month 6: 31.67, *p* < 0.001). The impact of OAB symptoms on negative emotions and embarrassment turned out to be significantly lower when the results of months 3 and 6 post treatment were compared with baseline scores. Meanwhile, no significant difference was found between the scores for months 3 and 6. The effect of botulinum toxin A injection on negative emotions associated with OAB was also analyzed in placebo-controlled studies. Chapple et al. [16] and Nitti et al. [17] observed a significant decrease in scores for the “Emotions” domain of the KHQ at 12 weeks post treatment, corresponding to an improvement in QoL in this area. In turn, Ginsberg et al. [28] reported a decrease in OAB-related depression resulting from repeated BoNT/A injections.

OAB symptoms may also trigger adaptive changes in the behavior of patients, who develop strategies to cope with the manifestation of the disease in public spaces; hence, adaptive behaviors without a doubt have a profound impact on one’s QoL. The “Undertaking Activities Mentioned in the Survey” domain of the ICIQ-LUTSqol refers to disruptions to everyday functioning due to such problems as wearing pads, changing underclothes frequently, reducing fluid intake, and being concerned about an unpleasant smell. The present study demonstrated a statistically significant post-treatment decrease in the impact of OAB symptoms on scores in the “Undertaking Activities Mentioned in the Survey” domain (mean baseline score: 73.08, score at month 3: 34.67, score at 6: 41.92, *p* < 0.001). According to Hamid et al. [15], the mean sum of supplies used by patients with OAB (pads/liners and diaper pants) decreased from a baseline level of 74.3 to 32.1 at 12 weeks post-treatment and remained at a similar level (27.2) up to week 52 after botulinum toxin injection. Tamburro et al. [21] also observed a substantial post-treatment decrease in the number of pads used (baseline: 2.4, week 12: 0.7). In a long-term study analyzing the effect of repeated treatment with botulinum toxin, Ginsberg et al. [28] found a decrease in the frequency of changing underclothes in 45–61% of the respondents. Questions included in the “Undertaking Activities Mentioned in the Survey” domain of the ICIQ-LUTSqol correspond well with some items included in the KHQ. A significant decrease in the impact of OAB symptoms on that area of life at 12 weeks after botulinum toxin injection was also observed by Chapple et al. [16] and Nitti et al. [17], who measured QoL with the KHQ. A decrease in UI episodes is associated with a lesser use of pads, which reflects a better QoL and diminished cost covered by the patients. Reduced use of pads and increased occupational activity are not only measures of botulinum toxin effectiveness in OAB but should also be considered positive economic indicators. Moreover, it should be emphasized that according to many authors, botulinum toxin injection itself is a potentially more cost-effective treatment option than oral pharmacotherapy [33,34,35].

### 3.3. Effect of Botulinum Toxin Treatment on QoL during the Management of Neurogenic Bladder

Botulinum toxin injections were also shown to be effective in managing neurogenic detrusor overactivity (NDO). In August 2011, botulinum toxin was approved by the US Food and Drug Agency (FDA) for the treatment of NDO associated with a concomitant neurological disorder in patients with multiple sclerosis (MS) and spinal cord injury (SCI) who did not respond adequately to anticholinergic agents or were intolerant to these drugs [36].

Both the American Urological Association (AUA) and the European Association of Urology (EAU) recommend botulinum toxin A therapy in patients with anticholinergic/antimuscarinic agent-resistant neurogenic bladder to reduce the frequency of incontinence episodes and improve QoL. In line with AUA guidelines, botulinum toxin A injections decreased the frequency of incontinence episodes compared with placebo regardless of the dose (200/300 units), and in most of the patients, the effectiveness of the treatment did not diminish with repeated injections of BoNT/A. According to the AUA, there is a dose-dependent risk of urinary retention requiring clean intermittent catheterization (CIC) [37,38]. However, some published studies analyzed the effectiveness of botulinum toxin injections at higher doses. For example, Nuanthaisong et al. [39] found that in women and men with neurogenic bladder, the treatment was effective when botulinum toxin was injected at 360 units. Injection of over 200 units of BoNT/A in patients with NDO is considered an off-label use, and in clinical trials, doses higher than 200 units are administered solely if, in the physician’s opinion, potential benefits outweigh the inherent risk [39].

A review of the available literature identified multiple studies confirming the effectiveness of botulinum toxin A injections in NDO, with a resultant improvement in patients’ QoL. In randomized placebo-controlled studies, botulin toxin A injections considerably reduced the frequency of UI episodes in women and men with NDO and contributed to an improvement in urodynamic parameters [40,41]. Importantly, a permanent attenuation of urinary symptoms after the treatment was associated with a clinically relevant improvement in the QoL of persons with OAB [42,43]. Interestingly, according to Khavari et al. [44], botulinum toxin injections in women with MS and neurogenic bladder appeared to increase activity in most brain areas involved in the perception and physiology of urinary urgency.

Recently, Walter et al. [45] presented the results of a phase IV trial documenting a beneficial effect of onabotulinum toxin injection on the occurrence of autonomic dysreflexia (AD) in patients with chronic SCI, with a simultaneous improvement in incontinence-related QoL.

Also, the results of meta-analyses point to the benefits of botulinum toxin treatment in women and men with NDO. According to Cheng et al. [46], the mean frequency of incontinence episodes in patients with NDO treated with onabotulinum toxin decreased significantly at week 6 post-injection as compared with placebo. Those findings are consistent with the results of other meta-analyses published by Wu et al. [47], Zhou et al. [48], and Yuan et al. [49]. Additionally, Cheng et al. [46] found no significant differences in the effectiveness of botulinum toxin administered at 200 and 300 units, which is in agreement with the results of meta-analyses published by Zhang et al. [50], Wu et al. [47], and Gu et al. [51]. Kennelly et al. [52] published the results of a long-term study analyzing the effectiveness and safety of botulinum toxin treatment during a four-year follow-up. The study showed that injections of onabotulinum toxin A reduced the frequency of UI episodes and improved the QoL in patients with NDO. Moreover, the authors did not observe an increase in the number of adverse events with repeated injections, which implies that BoNT/A treatment does not cause a toxic accumulation. Meanwhile, a meta-analysis conducted by Ni et al. [53] demonstrated that repeated botulinum toxin injections result in a permanent improvement in QoL in patients with NDO. In that meta-analysis, QoL improvement persisted in patients who received ≤4 repeated injections. However, as stated by the authors of that meta-analysis, more research is needed on the long-term effectiveness and safety of repeated botulinum toxin injections, especially in patients who require five injections or more. Notably, the SARS-CoV-2 pandemic has substantially disrupted regular treatment with botulinum toxin, giving researchers a unique opportunity to analyze the effects of delayed BoNT/A injection. A study conducted in Germany in a group of 100 patients with five various diseases requiring regular administration of BoNT/A demonstrated that a slight delay in botulinum toxin injection of a few weeks led to an exacerbation of symptoms and recurrence of underlying disease. According to the researchers, the return of patients’ status to the pre-pandemic level required continuous treatment lasting for at least one year [54].

Notably, botulinum toxin A is also tolerated well and is effective in managing NDO in children [55,56,57]. Additionally, some published evidence confirms that botulinum toxin is effective in treating non-neurogenic OAB in children and adolescents [58,59]. Meanwhile, Christiansen et al. [60] did not find a difference in the duration of the therapeutic effect of botulinum toxin in patients with idiopathic and neurogenic OAB. Published data also suggest that botulinum toxin may play a role in the management of interstitial cystitis and chronic bladder pain. Some studies demonstrated that intravesical botulinum toxin injections attenuated bladder pain, increased bladder volume, and subsided inflammation [61].

### 3.4. Exploratory Analysis

Analyzing the sociodemographic data of the study participants, we tried to explain whether their age, BMI, pregnancy, mode of delivery, and delivery of newborn with a birthweight ≥ 4 kg influenced the outcome of botulinum toxin treatment and QoL (Table A1, Table A2, Table A3, Table A4 and Table A5). However, given the potential limitations of the present study, especially the small sample size, all statistically significant relationships discussed below should be interpreted with caution.

According to the literature, higher BMI is a risk factor for OAB [62]. Indeed, our study participants presented with increased BMI (M = 28.47). In the study conducted by Miotła et al. [12], the mean BMI of the participants was 28.6 kg/m^2^. Similarly, participants of the study published by Tamburro et al. [21] showed a trend towards obesity. In the present study, the BMI and age of the study respondents did not correlate significantly with the treatment outcome and QoL (Table A1 and Table A2), which is in opposition to the results published by Hendrickson et al. [24]. According to those authors, in women with non-neurogenic UUI, older age and higher BMI were associated with a lesser reduction in the frequency of UUI episodes after administering BoNT/A at 100 or 200 units. Moreover, older age and higher BMI were shown to be associated with a shorter time till the recurrence of UUI. In a study conducted by Richter et al. [63], older age and higher BMI also correlated with a lesser mean reduction in the number of UUI episodes after treatment. Meanwhile, Komesu et al. [64] did not observe a significant association between an improvement in QoL after botulinum toxin treatment and the age of women with pharmacotherapy-resistant UUI. Finding a link between BMI and expected treatment effectiveness is paramount, given that excess body weight is a modifiable risk factor. According to Subak et al. [65], a reduction in body weight after bariatric treatment was associated with a considerable decrease in the occurrence of UI in female and male patients during a three-year follow-up. Thus, a question arises whether a reduction in body weight before OAB treatment could improve the effectiveness of the therapy.

To the best of our knowledge, none of the previously published studies analyzed the effects of pregnancy, mode of delivery, and delivering a baby with a birth weight ≥ 4 kg on the outcomes of botulinum toxin treatment in women with OAB. Our present study showed no significant associations between pregnancy, treatment outcomes, and post-treatment improvement in QoL (Table A3, Table A4 and Table A5). However, our analysis of a link between delivering a baby with a birth weight ≥ 4 kg and potential lesser improvement in QoL after botulinum toxin treatment might have been biased, given there was only a slight difference between the examined groups. Notably, our present study only included women, and according to Hsiao et al. [66], the female sex is associated with better outcomes of botulinum toxin treatment in OAB. Abrar et al. [67] analyzed the predictors of no response of OAB patients to botulinum toxin treatment. The identified factors included older age, tobacco smoking, and various urodynamic parameters. However, as emphasized by the authors, the results of their study were preliminary and need to be verified in future. The same study demonstrated that women with a history of hysterectomy were at increased risk of CIC after BoNT/A injection. History of previous urogynecological surgery was an exclusion criterion in the present study. In turn, Majoros et al. [68] did not identify any perioperative factor that would significantly influence the outcome of botulinum toxin treatment, namely success and complication rates.

It is worth emphasizing that intravesical botulinum toxin injections may also pose a risk of adverse effects. In the study by Hamid et al. [15], adverse events, such as urinary retention, micturition problems, and urinary tract infections, occurred in 2.6% of patients. In the present study, urinary tract infections were observed in four patients (4%) and one person (1%) required CIC. Urinary retention, a potential adverse event after intravesical injection of botulinum toxin, requires frequent intermittent catheterization, which may discourage patients from this treatment method. However, according to the literature, the prevalence of urinary retention that requires CIC is low. In a retrospective study of 99 patients, the prevalence of urinary retention requiring CIC after the intravesical injection of 100 units of botulinum toxin for OAB treatment was 1.6% [69]. In another study by Miotła et al. [12], CIC was required in 2 out of 76 patients. According to Rechberger et al. [11], CIC was needed in 1 out of 10 female patients, and despite that complication, the patient that needed it would still recommend botulinum toxin injections to other individuals with OAB. The authors of another study, Miotła et al. [70], demonstrated that women who required CIC after botulinum toxin injection presented with higher-risk pregnancy and more often than other patients had a history of vaginal delivery. Tamburro et al. [21] reported no cases of urinary tract infections, and CIC, lasting for a few weeks, was required in merely 2 out of 22 patients. Meanwhile, the results of randomized studies suggest that CIC was needed in 6–6.9% of women after botulinum toxin injection [16,17,18]. A long-term study conducted by Nitti et al. [71] in a group of female and male patients demonstrated that the risk of CIC during subsequent treatment courses was lower in persons who did not need to perform self-catheterization after the first botulinum toxin injection. Finally, according to Fowler et al. [27], at 36 weeks after botulinum toxin treatment, the improvement in HR-QoL of patients who required CIC was similar to that in those who did not need the catheterization. A study conducted by Mühlstädt et al. [72] showed that up to 66% of patients receiving botulinum toxin injections for OAB would choose this treatment modality once again. Hendrickson et al. [24] demonstrated that in women with non-neurogenic UUI, a higher risk of CIC correlated with lower BMI, pre-menopausal status, and administration of BoNT/A at 200 units. A meta-analysis published by Ramos et al. [73] revealed that patients receiving 100 units of botulinum toxin did not differ from those receiving higher doses in terms of the incidence of urinary retention. Published evidence shows that antibiotic prophylaxis may reduce the risk of urinary tract infection after botulinum toxin treatment [74]. Importantly, antibiotic prophylaxis was used in all patients included in the present study, but, as mentioned above, four of them (4%) developed urinary tract infections. In the study conducted by Hamid et al. [15], urinary tract infections occurred in 0.4% of all patients, although, as mentioned by the authors, not all of them received prophylactic antibiotic therapy during botulinum toxin treatment. Prophylactic antibiotics were administered in 69% of patients participating in that study, including those who eventually developed a urinary tract infection. In a randomized study by Nitti et al. [17], urinary tract infections occurred in 15% of persons receiving botulinum toxin injections. One strength of the present study stems from the fact that intravesical injections of botulinum toxin A in all patients were performed at one center, by the same physician.

## 4. Conclusions

The QoL of the study participants improved at three and six months after botulinum toxin treatment;At six months post-treatment, OAB symptoms were more severe than at three months, and their impact on QoL increased;Botulinum toxin is effective in the treatment of OAB.

## 5. Materials and Methods

### 5.1. Methods and Research Instruments

The QoL of the study participants before and after botulinum toxin treatment was evaluated with a survey developed by the authors solely for the present study and several validated scales: ICIQ-OAB (International Consultation on Incontinence Questionnaire Overactive Bladder Module), ICIQ-OABqol (International Consultation on Incontinence Questionnaire Overactive Bladder Quality of Life Module), and ICIQ-LUTSqol (International Consultation on Incontinence Questionnaire Lower Urinary Tract Symptoms Quality of Life Module). To facilitate the interpretation of the results, the scores for each scale were summed up and converted onto a scale from 0 to 100, whereby lower scores corresponded to a better QoL [75].

The survey developed by the authors was designed to obtain the sociodemographic characteristics of the participants and their selected medical data. The survey consisted of three parts. The first part contained questions about the basic sociographic characteristics of the respondents. The second part referred to the urogynecological history of the respondents, including the time of the first OAB manifestations, the time when OAB treatment was implemented, familial history of urinary bladder disorders, and gynecological history. The third part contained questions about the obstetrical history of the studied women, such as pregnancy, parity, mode of delivery, and birth weight.

Polish versions of the ICIQ-OAB, ICIQ-OABqol, and ICIQ-LUTSqol questionnaires were obtained from the iciq.net website. For the present study, questions of the ICIQ-OABqol instrument were grouped into four domains: “Copying”, “Concern”, “Sleep”, and “Social Interactions” (Table 17). According to Coyne et al. [76], the four-factor system is the most accurate and suitable interpretation. Questions of the ICIQ-LUTSqol questionnaire were grouped into domains (Table 18) according to Hebber et al. [77], based on the KHQ.

### 5.2. Organization and Protocol of the Study

The protocol of the study was approved by the Local Bioethics Committee at the Pomeranian Medical University in Szczecin (decision no. KB-0012/125/17 of 16 November 2017). The study was carried out between November 2017 and September 2021 in a group of women with OAB recruited at the Clinic of Gynecology, Endocrinology, and Gynecologic Oncology, Public Clinical Hospital No. 1, Pomeranian Medical University in Szczecin.

Each patient who satisfied the inclusion criteria was provided with general information about the study, including its topic, objectives, protocol, and survey instructions. After expressing written informed consent to participate in the study, the patients received the ICIQ-OAB, ICIQ-OABqol, and ICIQ-LUTSqol questionnaires and the survey developed by the authors. The survey was filled in only once before botulinum toxin treatment. The respondents were informed that participation in the study was entirely voluntary and anonymous, and data collected during the study would be used solely for research purposes and would have no impact on the treatment process. Moreover, the respondents were informed that they could withdraw from the study at any stage without specifying a reason. Three and six months after the treatment, the respondents were again asked to fill in the validated scales. The study procedures were designed to be conducted during routine follow-up visits at the Clinic of Gynecology, Endocrinology, and Gynecologic Oncology. However, given the COVID-19 pandemic and resultant cancellations of scheduled visits, 42 respondents answered the questions included in the questionnaires via a telephone survey.

### 5.3. Intravesical Injection of Botulinum Toxin A

All patients with pharmacotherapy-resistant OAB received 100 units of botulinum toxin A (Botox^®^, Allergan Inc., Irvine, CA, USA) in 10 mL physiological saline. Botulinum toxin A was administered as multiple, evenly distributed injections to the bladder’s muscular layer, delivered at 20 spots (5 units in 0.5 mL per spot), omitting the trigonum vesicae. The injections were delivered with a rigid cystoscope, under local anesthesia with short-time general anesthesia or without. The procedure was always carried out by the same operator. All patients received prophylactic antibiotics (Fosfomycin) one day before and after the procedure, as well as on the day of the procedure. The patients were discharged home on the day of the procedure after confirming that they could pass clear urine. Urine retention was defined as the post-void residual (PVR) ≥350 mL regardless of symptoms or PVR between ≥200 and <350 mL with accompanying symptoms that required CIC in the operator’s opinion. Urinary tract infection was defined as obtaining a positive result of urine culture within the first week of the treatment.

### 5.4. Characteristics of the Study Group

Out of 124 patients with a history of OAB diagnosis, 100 women who satisfied the inclusion criteria listed in Table 19 were eventually enrolled.

The result of urodynamic testing was not considered an inclusion criterion of this study. This study exclusively included women who received an intravesical injection of botulinum toxin for the first time.

QoL after botulinum toxin injection was evaluated in women aged from 30 to 87 years (mean 58.82 years). The mean BMI of the study respondents was 28.47 kg/m^2^. Most of the respondents lived in cities/towns (82%), had at least secondary education (42%), were employed as white-collar workers currently or in the past (63%), or were retired (51%).

The second part of the survey included questions about the urogynecological history of the respondents. Based on the survey data, the mean time elapsed since the first manifestation of OAB symptoms was estimated at 11 years, and the mean duration of anti-OAB treatment was five years. The mean time between the first manifestation of OAB and the implementation of treatment was six years. Up to 70% of the respondents did not report a familial history of urinary bladder disorders or were unaware of such a history. Another 30% of the study participants presented with a familial history of urinary bladder problems, with the most often affected member being their mother. The last question included in this part of the survey referred to the gynecological history of the respondents and was aimed at increasing the homogeneity of the study group.

The third part of the survey referred to obstetrical history, including pregnancy and delivery mode. Up to 85.71% and 14.29% of all deliveries were vaginal and cesarean births, respectively. During vaginal births, episiotomy was needed in 72.67% of the cases, and perineal rupture occurred in 21.33%. Instrumental deliveries occurred in three cases, with vacuum extractors used in two cases and forceps in one. Birth weight ≥ 4 kg was recorded in 8.57% of the newborns.

Four respondents (4%) developed urinary tract infections after treatment with botulinum toxin A, and one woman (1%) required CIC due to urinary retention.

## Figures and Tables

**Table 1 toxins-16-00007-t001:** Basic descriptive statistics along with the results of the Shapiro–Wilk normality test for the ICIQ-OAB scores.

Variable	M	Me	SD	Sk	Curt	Min	Max	W	*p*
How often do you pass urine during the day? [P1]	69.75	75.00	26.42	−0.41	−1.04	25.00	100.00	0.85	<0.001
How often do you pass urine during the day? [P2]	20.00	0.00	25.87	1.31	1.22	0.00	100.00	0.76	<0.001
How often do you pass urine during the day? [P3]	29.50	25.00	27.38	0.72	−0.25	0.00	100.00	0.86	<0.001
During the night, how many times do you have to get up to urinate, on average? [P1]	75.75	75.00	23.96	−0.69	−0.48	25.00	100.00	0.83	<0.001
During the night, how many times do you have to get up to urinate, on average? [P2]	34.25	25.00	31.51	0.62	−0.73	0.00	100.00	0.86	<0.001
During the night, how many times do you have to get up to urinate, on average? [P3]	41.50	25.00	29.56	0.51	−0.55	0.00	100.00	0.89	<0.001
Do you have to rush to the toilet to urinate? [P1]	85.00	100.00	20.10	−1.33	1.93	0.00	100.00	0.72	<0.001
Do you have to rush to the toilet to urinate? [P2]	31.50	25.00	26.74	0.62	−0.27	0.00	100.00	0.88	<0.001
Do you have to rush to the toilet to urinate? [P3]	40.25	25.00	28.62	0.44	−0.59	0.00	100.00	0.90	<0.001
Does urine leak before you can get to the toilet? [P1]	63.25	75.00	25.49	−0.70	0.42	0.00	100.00	0.87	<0.001
Does urine leak before you can get to the toilet? [P2]	21.75	25.00	23.48	0.94	0.37	0.00	100.00	0.81	<0.001
Does urine leak before you can get to the toilet? [P3]	28.75	25.00	26.20	0.55	−0.65	0.00	100.00	0.86	<0.001
Overall score [P1]	293.75	300.00	60.76	−0.24	−0.60	150.00	400.00	0.97	0.011
Overall score [P2]	107.50	75.00	88.66	0.67	−0.46	0.00	350.00	0.92	<0.001
Overall score [P3]	140.00	125.00	88.83	0.45	−0.57	0.00	350.00	0.96	0.001

M—mean; Me—median; SD—standard deviation; Sk—skewness; Curt—kurtosis; Min—minimum value; Max—maximum value; W—Shapiro–Wilk statistic; *p*—significance level; P1—measurement 1; P2—measurement 2; P3—measurement 3.

**Table 2 toxins-16-00007-t002:** Basic descriptive statistics along with the results of the Shapiro–Wilk normality test for the ICIQ-OABqol scores.

Variable	M	Me	SD	Sk	Curt	Min	Max	W	*p*
Copying [P1]	72.73	76.25	21.33	−0.66	−0.43	15.00	100.00	0.93	<0.001
Copying [P2]	27.58	20.00	27.15	0.86	−0.34	0.00	92.50	0.87	<0.001
Copying [P3]	33.85	25.00	29.07	0.54	−1.07	0.00	90.00	0.90	<0.001
Concern [P1]	65.86	68.57	21.76	−0.48	−0.43	8.57	100.00	0.96	0.007
Concern [P2]	23.06	15.71	23.37	0.89	−0.26	0.00	82.86	0.87	<0.001
Concern [P3]	29.23	22.86	26.80	0.50	−1.09	0.00	82.86	0.89	<0.001
Sleep [P1]	65.32	72.00	23.95	−0.65	−0.43	4.00	100.00	0.94	<0.001
Sleep [P2]	25.96	16.00	26.44	0.95	−0.13	0.00	100.00	0.87	<0.001
Sleep [P3]	32.92	24.00	27.98	0.66	−0.77	0.00	100.00	0.90	<0.001
Social interaction [P1]	42.88	44.00	24.80	0.04	−0.96	0.00	92.00	0.97	0.015
Social interaction [P2]	14.48	4.00	19.86	1.32	0.66	0.00	72.00	0.76	<0.001
Social interaction [P3]	17.32	4.00	21.88	1.06	−0.19	0.00	72.00	0.78	<0.001
Overall score [P1]	246.78	247.39	79.51	−0.34	−0.77	56.14	380.00	0.97	0.016
Overall score [P2]	91.07	58.61	92.98	0.99	−0.10	0.00	327.86	0.86	<0.001
Overall score [P3]	113.32	78.68	100.61	0.64	−0.94	0.00	327.86	0.89	<0.001

M—mean; Me—median; SD—standard deviation; Sk—skewness; Curt—kurtosis; Min—minimum value; Max—maximum value; W—Shapiro–Wilk statistic; *p*—significance level; P1—measurement 1; P2—measurement 2; P3—measurement 3.

**Table 3 toxins-16-00007-t003:** Basic descriptive statistics along with the results of the Shapiro–Wilk normality test for the ICIQ-LUTSqol scores.

Variable	M	Me	SD	Sk	Curt	Min	Max	W	*p*
Role limitations [P1]	80.50	83.33	21.72	−0.80	−0.59	33.33	100.00	0.81	<0.001
Role limitations [P2]	29.33	33.33	30.53	0.84	−0.33	0.00	100.00	0.84	<0.001
Role limitations [P3]	37.67	33.33	29.83	0.42	−0.94	0.00	100.00	0.91	<0.001
Physical limitations [P1]	83.50	100.00	22.54	−1.37	0.92	16.67	100.00	0.74	<0.001
Physical limitations [P2]	31.50	33.33	32.12	0.82	−0.47	0.00	100.00	0.85	<0.001
Physical limitations [P3]	39.33	33.33	31.74	0.39	−1.06	0.00	100.00	0.90	<0.001
Social limitations [P1]	54.83	66.67	32.85	−0.16	−1.03	0.00	100.00	0.91	<0.001
Social limitations [P2]	16.83	0.00	26.43	1.55	1.58	0.00	100.00	0.68	<0.001
Social limitations [P3]	21.00	0.00	26.44	1.00	−0.12	0.00	100.00	0.78	<0.001
Personal relationships [P1]	51.36	55.56	34.76	−0.15	−1.28	0.00	100.00	0.91	<0.001
Personal relationships [P2]	16.17	0.00	25.54	1.31	0.28	0.00	88.89	0.67	<0.001
Personal relationships [P3]	21.06	0.00	28.01	1.13	0.07	0.00	100.00	0.76	<0.001
Emotions [P1]	60.44	55.56	26.11	−0.10	−0.91	0.00	100.00	0.95	<0.001
Emotions [P2]	20.89	11.11	25.43	1.17	0.46	0.00	88.89	0.80	<0.001
Emotions [P3]	27.67	22.22	27.28	0.72	−0.55	0.00	100.00	0.87	<0.001
Sleep/Energy [P1]	69.83	66.67	25.15	−0.53	−0.58	0.00	100.00	0.91	<0.001
Sleep/Energy [P2]	28.83	33.33	27.81	0.84	0.02	0.00	100.00	0.87	<0.001
Sleep/Energy [P3]	37.33	33.33	28.34	0.60	−0.48	0.00	100.00	0.91	<0.001
Undertaking activities mentioned in the survey [P1]	73.08	79.17	23.18	−1.14	1.16	0.00	100.00	0.89	<0.001
Undertaking activities mentioned in the survey [P2]	34.67	29.17	28.81	0.53	−0.76	0.00	100.00	0.92	<0.001
Undertaking activities mentioned in the survey [P3]	41.92	41.67	30.53	0.22	−1.02	0.00	100.00	0.94	<0.001
Embarrassment [P1]	67.33	66.67	32.82	−0.62	−0.71	0.00	100.00	0.83	<0.001
Embarrassment [P2]	25.00	33.33	27.78	0.93	0.22	0.00	100.00	0.79	<0.001
Embarrassment [P3]	31.67	33.33	32.26	0.72	−0.50	0.00	100.00	0.82	<0.001
Overall score [P1]	535.75	538.89	157.46	−0.15	−0.88	219.44	800.00	0.97	0.028
Overall score [P2]	201.61	141.67	195.82	0.95	−0.10	0.00	722.22	0.88	<0.001
Overall score [P3]	254.69	198.61	202.56	0.53	−0.91	0.00	736.11	0.92	<0.001

M—mean; Me—median; SD—standard deviation; Sk—skewness; Curt—kurtosis; Min—minimum value; Max—maximum value; W—Shapiro–Wilk statistic; *p*—significance level; P1—measurement 1; P2—measurement 2; P3—measurement 3.

**Table 4 toxins-16-00007-t004:** Basic descriptive statistics along with the results of the Shapiro–Wilk normality test for the impact of OAB symptoms on everyday life (ICIQ-LUTSqol).

Variable	M	Me	SD	Sk	Curt	Min	Max	W	*p*
Impact on everyday life [P1]	9.09	10.00	1.24	−1.50	2.00	5.00	10.00	0.74	<0.001
Impact on everyday life [P2]	3.00	2.00	3.03	0.83	−0.58	0.00	10.00	0.84	<0.001
Impact on everyday life [P3]	3.98	3.00	3.33	0.52	−1.19	0.00	10.00	0.86	<0.001

M—mean; Me—median; SD—standard deviation; Sk—skewness; Curt—kurtosis; Min—minimum value; Max—maximum value; W—Shapiro–Wilk statistic; *p*—significance level; P1—measurement 1; P2—measurement 2; P3—measurement 3.

**Table 5 toxins-16-00007-t005:** Results of repeated measures analysis of variance for variables derived from the ICIQ-OAB questionnaire.

Variable	df1	df2	F	*p*	η^2^
How often do you pass urine during the day?	1.74	161.76	194.79	<0.001	0.66
During the night, how many times do you have to get up to urinate, on average?	1.55	153.47	138.83	<0.001	0.58
Do you have to rush to the toilet to urinate?	1.52	150.58	226.62	<0.001	0.70
Does urine leak before you can get to the toilet?	1.64	162.20	159.47	<0.001	0.62
Overall score	1.61	159.65	28,815	<0.001	0.74

df1, df2—degrees of freedom; F—Fisher statistic for ANOVA; *p*—significance level; η^2^—effect size.

**Table 6 toxins-16-00007-t006:** Analysis of differences between three measurements taken with the ICIQ-OAB questionnaire; results of repeated measures ANOVA.

Variable	Measurement 1		Measurement 2		Measurement 3	
	M	SD	M	SD	M	SD
How often do you pass urine during the day?	69.75 ^ab^	26.42	20.00 ^ac^	25.87	29.50 ^bc^	27.38
During the night, how many times do you have to get up to urinate, on average?	75.75 ^ab^	23.96	34.25 ^ac^	31.51	41.50 ^bc^	29.56
Do you have to rush to the toilet to urinate?	85.00 ^ab^	20.10	31.50 ^ac^	26.74	40.25 ^bc^	28.62
Does urine leak before you can get to the toilet?	63.25 ^ab^	25.49	21.75 ^ac^	23.48	28.75 ^bc^	26.20
Overall score	293.75 ^ab^	60.76	107.50 ^ac^	88.66	140.00 ^bc^	88.83

^a,b,c^ statistically significant (*p* < 0.05) result of the post hoc test with the Sidak correction; M—mean; SD—standard deviation.

**Table 7 toxins-16-00007-t007:** Results of repeated measures analysis of variance for variables derived from the ICIQ-OABqol questionnaire.

Variable	df1	df2	F	*p*	η^2^
Copying	1.70	168.46	188.82	<0.001	0.66
Concern	1.58	156.39	183.72	<0.001	0.65
Sleep	1.72	170.23	141.00	<0.001	0.59
Social interaction	1.45	143.89	130.93	<0.001	0.57
Overall score	1.66	164.68	194.70	<0.001	0.66

df1, df2—degrees of freedom; F—Fisher statistic for ANOVA; *p*—significance level; η^2^—effect size.

**Table 8 toxins-16-00007-t008:** Analysis of differences between three measurements taken with the ICIQ-OABqol questionnaire; results of repeated measures ANOVA.

Variable	Measurement 1		Measurement 2		Measurement 3	
	M	SD	M	SD	M	SD
Copying	72.73 ^ab^	21.33	27.58 ^ac^	20.00	33.85 ^bc^	29.07
Concern	65.86 ^ab^	21.76	23.06 ^ac^	15.71	29.23 ^bc^	26.80
Sleep	65.32 ^ab^	23.95	25.96 ^ac^	16.00	32.92 ^bc^	27.98
Social interaction	42.88 ^ab^	24.80	14.48 ^a^	4.00	17.32 ^b^	21.88
Overall score	246.78 ^ab^	79.51	91.07 ^ac^	58.61	113.32 ^bc^	100.61

^a,b,c^ statistically significant (*p* < 0.05) result of the post hoc test with the Sidak correction; M—mean; SD—standard deviation.

**Table 9 toxins-16-00007-t009:** Results of repeated measures analysis of variance for variables derived from the ICIQ-LUTSqol questionnaire.

Variable	df1	df2	F	*p*	η^2^
Role limitations	1.75	173.62	165.47	<0.001	0.63
Physical limitations	1.77	175.45	161.27	<0.001	0.62
Social limitations	1.61	159.73	94.79	<0.001	0.49
Personal relationships	1.28	109.00	78.95	<0.001	0.48
Emotions	1.63	161.43	114.94	<0.001	0.54
Sleep/Energy	1.50	148.22	129.94	<0.001	0.57
Undertaking activities mentioned in the survey	1.66	163.95	126.18	<0.001	0.56
Embarrassment	1.71	169.58	117.26	<0.001	0.54
Overall score	1.70	167.87	193.78	<0.001	0.66

df1, df2—degrees of freedom; F—Fisher statistic for ANOVA; *p*—significance level; η^2^—effect size.

**Table 10 toxins-16-00007-t010:** Analysis of differences between three measurements taken with the ICIQ-LUTSqol questionnaire; results of repeated measures ANOVA.

Variable	Measurement 1		Measurement 2		Measurement 3	
	M	SD	M	SD	M	SD
Role limitations	80.50 ^ab^	21.72	29.33 ^ac^	30.53	37.67 ^bc^	29.83
Physical limitations	83.50 ^ab^	22.54	31.50 ^ac^	32.12	39.33 ^bc^	31.74
Social limitations	54.83 ^ab^	32.85	16.83 ^a^	26.43	21.00 ^b^	26.44
Personal relationships	51.36 ^ab^	34.76	16.17 ^ac^	25.54	21.06 ^bc^	28.01
Emotions	60.44 ^ab^	26.11	20.89 ^ac^	25.43	27.67 ^bc^	27.28
Sleep/Energy	69.83 ^ab^	25.15	28.83 ^ac^	27.81	37.33 ^bc^	28.34
Undertaking activities mentioned in the survey	73.08 ^ab^	23.18	34.67 ^ac^	28.81	41.92 ^bc^	30.53
Embarrassment	67.33 ^ab^	32.82	25.00 ^ac^	27.78	31.67 ^bc^	32.26
Overall score	535.75 ^ab^	157.46	201.61 ^ac^	195.82	254.69 ^bc^	202.56

^a,b,c^ statistically significant (*p* < 0.05) result of the post hoc test with Sidak correction; M—mean; SD—standard deviation.

**Table 11 toxins-16-00007-t011:** Results of repeated measures analysis of variance for the impact of OAB symptoms on everyday life.

Variable	df1	df2	F	*p*	η^2^
Impact on everyday life	1	99	640.04	<0.001	0.71

df1, df2—degrees of freedom; F—Fisher statistic for ANOVA; *p*—significance level; η^2^—effect size.

**Table 12 toxins-16-00007-t012:** Analysis of differences between the three measurements of OAB symptom effect on everyday life; results of repeated measures ANOVA.

	Measurement 1		Measurement 2		Measurement 3	
Variable	M	SD	M	SD	M	SD
Impact on everyday life	9.09 ^ab^	1.24	3.00 ^ac^	0.83	3.98 ^bc^	3.33

^a,b,c^ statistically significant (*p* < 0.05) result of the post hoc test with the Sidak correction; M—mean; SD—standard deviation.

**Table 13 toxins-16-00007-t013:** Results of Friedman ANOVA for the impact of OAB symptoms.

Variable	df	Χ^2^	*p*	Kendall W
How often do you pass urine during the day?	2	145.39	<0.001	2.91
During the night, how many times do you have to get up to urinate, on average?	2	133.03	<0.001	2.66
Do you have to rush to the toilet to urinate?	2	147.91	<0.001	2.96
Does urine leak before you can get to the toilet?	2	141.43	<0.001	2.83

df—degrees of freedom; X^2^—chi-square statistic; *p*—significance level; Kendall W—Kendall’s W-coefficient of concordance.

**Table 14 toxins-16-00007-t014:** Mean ranks for Friedman ANOVA, along with the results of post hoc tests for the impact of OAB symptoms.

Variable	Measurement 1	Measurement 2	Measurement 3
	M_rank_	M_rank_	M_rank_
How often do you pass urine during the day?	2.89 ^ab^	1.46 ^a^	1.66 ^b^
During the night, how many times do you have to get up to urinate, on average?	2.83 ^ab^	1.47 ^a^	1.71 ^b^
Do you have to rush to the toilet to urinate?	2.88 ^ab^	1.41 ^a^	1.72 ^b^
Does urine leak before you can get to the toilet?	2.84 ^ab^	1.48 ^a^	1.68 ^b^

^a**,**b^—statistically significant (*p* < 0.05) results of post hoc tests with the Bonferroni correction.

**Table 15 toxins-16-00007-t015:** Results of Friedman ANOVA for the impact of OAB on various aspects of life.

Variable	df	Χ^2^	*p*	Kendall W
Household tasks	2	135.50	<0.001	2.70
Normal daily activities outside the home	2	144.61	<0.001	2.89
Physical activities	2	136.00	<0.001	2.72
Ability to travel	2	131.37	<0.001	2.63
Social life	2	120.86	<0.001	2.42
Ability to see/visit friends	2	110.21	<0.001	2.20
Relationship with partner	2	64.41	<0.001	1.29
Sex life	2	55.11	<0.001	1.10
Family life	2	94.57	<0.001	1.89
Feeling depressed	2	107.16	<0.001	2.14
Feeling anxious or nervous	2	125.91	<0.001	2.52
Feeling bad about herself	2	84.07	<0.001	1.68
Sleep quality	2	123.56	<0.001	2.47
Feeling worn out/tired	2	128.58	<0.001	2.57
Wearing pads	2	109.11	<0.001	2.18
Controlling fluid intake	2	102.17	<0.001	2.04
Changing underclothes when they get wet	2	121.84	<0.001	2.44
Worrying because of smell	2	118.55	<0.001	2.37
Getting embarrassed	2	120.35	<0.001	2.41

df—degrees of freedom; X^2^—chi-square statistic; *p*—significance level; Kendall W—Kendall’s W-coefficient of concordance.

**Table 16 toxins-16-00007-t016:** Mean ranks for Friedman ANOVA, along with the results of post hoc tests for the impact of OAB on various aspects of life.

Variable	Measurement 1	Measurement 2	Measurement 3
	M_rank_	M_rank_	M_rank_
Household tasks	2.86 ^ab^	1.47 ^a^	1.68 ^b^
Normal daily activities outside the home	2.88 ^ab^	1.45 ^a^	1.67 ^b^
Physical activities	2.85 ^ab^	1.49 ^a^	1.66 ^b^
Ability to travel	2.83 ^ab^	1.48 ^a^	1.70 ^b^
Social life	2.76 ^ab^	1.55 ^a^	1.69 ^b^
Ability to see/visit friends	2.71 ^ab^	1.59 ^a^	1.71 ^b^
Relationship with partner	2.43 ^ab^	1.73 ^a^	1.84 ^b^
Sex life	2.39 ^ab^	1.74 ^a^	1.88 ^b^
Family life	2.58 ^ab^	1.66 ^a^	1.77 ^b^
Feeling depressed	2.74 ^ab^	1.55 ^a^	1.72 ^b^
Feeling anxious or nervous	2.82 ^ab^	1.46 ^a^	1.73 ^b^
Feeling bad about herself	2.58 ^ab^	1.60 ^a^	1.83 ^b^
Sleep quality	2.79 ^ab^	1.45 ^ac^	1.67 ^bc^
Feeling worn out/tired	2.81 ^ab^	1.49 ^a^	1.70 ^b^
Wearing pads	2.70 ^ab^	1.53 ^a^	1.77 ^b^
Controlling fluid intake	2.70 ^ab^	1.52 ^a^	1.78 ^b^
Changing underclothes when they get wet	2.76 ^ab^	1.48 ^a^	1.77 ^b^
Worrying because of smell	2.75 ^ab^	1.53 ^a^	1.73 ^b^
Getting embarrassed	2.76 ^ab^	1.51 ^a^	1.73 ^b^

^a**,**b**,**c^—statistically significant (*p* < 0.05) results of post hoc tests with the Bonferroni correction.

**Table 17 toxins-16-00007-t017:** Domains of the ICIQ-OABqol questionnaire, along with respective questions.

Domain Name	Questions	No. of Questions per Domain
Copying	3, 5, 10, 15,16, 20, 26, 27	8
Concern	6, 7, 8, 13, 17, 19, 23	7
Sleep	4, 9, 11, 18, 24	5
Social Interactions	12, 14, 21, 22, 25	5

**Table 18 toxins-16-00007-t018:** Domains of the ICIQ-LUTSqol questionnaire, along with respective questions assigned based on the KHQ.

Name of the KHQ Domain	KHQ Questions	Corresponding ICIQ-LUTSqol Questions
Role Limitations	3a, 3b	3a, 4a
Physical Limitation	4a, 4b	5a, 6a
Social Limitation	4c, 4d	7a, 8a
Personal Relationships	5a, 5b, 5c	9a, 10a, 11a
Emotions	6a, 6b, 6c	12a, 13a, 14a
Sleep/Energy	7a, 7b	15a, 16a,
Undertaking activities mentioned in the survey	8a, 8b, 8c, 8d	17a,18a,19a, 20a

**Table 19 toxins-16-00007-t019:** Inclusion and exclusion criteria of the study.

Inclusion Criteria	Exclusion Criteria
intolerance or unresponsiveness to pharmacotherapy with anticholinergic agents and/or β3 agonists	history of previous treatment with botulinum toxin because of other diseases/indications (within six months before the enrollment)
diagnosis of OAB	urinary tract infection
written informed consent to participate	predominant component of stress incontinence
age ≥ 18 years	female pelvic organ prolapse ≥ 2°
	history of a recent urogynecological surgery (all surgeries used to treat stress urinary incontinence and female pelvic organ prolapse)

## Data Availability

The data presented in this study are openly available in: Licow, A.; Ciećwież, S.; Brodowska. 2022. Quality of life in patients with overactive bladder following botulinum toxin treatment: a preliminary report; Ginekol. Pol.; doi:10.5603/GP.a2022.0105.

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
