# Peer review of "Quality of Life in Female Patients with Overactive Bladder after Botulinum Toxin Treatment"

_toxins, 2023, doi:10.3390/toxins16010007_

Round 1

Reviewer 1 Report

Comments and Suggestions for Authors

This study evaluated the quality of life in women with overactive bladder before the injection of botulinum toxin and three and six months after the intervention.

This study provides further evidence of the QoL in patients with OAB. The manuscript is verbose mainly in the introduction and in the results description.

The results are reliable, and the statistical analysis is well structured. It is not clear why they used a “diagnostic survey” as study design: it looks like a prospective cohort study. The conclusions are detailed.  The study is clinically relevant showing the improvement of the QoL in women with OAB treated with botulinum toxin.

No typos detected.

The title of manuscript is descriptive with a right length. Study population is not clear. It could be better to modify it as follows: “Quality of life in female patients with overactive bladder after botulinum toxin treatment”.

The abstract follows the style requested by the journal.

The introduction highlights the current state of the research field and defines the purpose/goals of the study. But It is too long and redundant.

The purpose of the study is clearly reported.  However, the study design should be better defined as well as the “additional questionnaire” cited in the abstract. The cohort analyzed is adequate and the data collection is comprehensive. The variables are properly described. Statistical analysis is correct and reported.

The results are correctly reported. Tables are difficult to read. E.g.: for the Table 1-4 I would suggest using a different format, making the variables easier to visualize and comparisons to be much more comprehensible.

The discussion is too long (more than 4-5 paragraphs) therefore it’s suggested to merge some sections. Despite the length, the main clinical message is clearly reported, and the findings have been compared with the literature. Future perspectives and limits of the study are described in the discussion section.

I recommend a re-editing of the tables to make easier the interpretation.

References are comprehensive, correctly numbered and formatted as requested .

Author Response

Dear reviewer,

In the attachment I am sending responses to comments and suggestions regarding the article. I hope that my answers to your questions and the corrections included in the article will be satisfactory.

Kind regards.

Reviewer 2 Report

Comments and Suggestions for Authors

This manuscript delves into the assessment of the quality of life in patients with Overactive Bladder (OAB) following intravesical botulinum toxin injection. While the content appears intriguing, some aspects require clarification. Specifically, the rationale behind utilizing intravesical botulinum toxin therapy should be restructured.

Introduction

1.     On page 2, lines 64-70, the elucidation of botulinum toxin mechanisms lacks precision, and the cited reference [7] is outdated. It is recommended to refer to Hu JC et al's recent work (Toxins 2023, 15, 166) for an updated understanding.

Discussion

1.     On page 11, lines 229-230, the rationale for OAB treatment needs reorganization. Please consult Lin CT et al's work (Uro Sci 2020; 31: 91-98) and the American Urological Association (AUA) guidelines for a comprehensive perspective.

2.     On page 13, lines 312-315, it is essential to acknowledge that different botulinum toxin products may yield varying efficacy, and their dosages are not directly interchangeable. The authors should explicitly address this concept to enhance clarity.

Comments on the Quality of English Language

The sentence must be more precise and concise. 

Author Response

(The authors gave the same response as above.)
